# Biocomposite Coatings Delay Senescence in Stored *Diospyros kaki* Fruits by Regulating Antioxidant Defence Mechanism and Delaying Cell Wall Degradation

Muhammad Shahzad Saleem [1], Shaghef Ejaz [1,*], Walid F. A. Mosa [2,*], Sajid Ali [1], Hasan Sardar [1], Muhammad Moaaz Ali [3], Sami Ullah [4], Hayssam M. Ali [5], Anna Lisek [6] and Muhammad Akbar Anjum [1]

1 Department of Horticulture, Bahauddin Zakariya University, Multan 60800, Pakistan
2 Plant Production Department (Horticulture-Pomology), Faculty of Agriculture, Saba Basha, Alexandria University, Alexandria 21531, Egypt
3 College of Horticulture, Fujian Agriculture and Forestry University, Fuzhou 350002, China
4 Department of Horticulture, Muhammad Nawaz Sharif University of Agriculture, Multan 60000, Pakistan
5 Department of Botany and Microbiology, College of Science, King Saud University, Riyadh 11451, Saudi Arabia
6 The National Institute of Horticultural Research, Konstytucji 3 Maja 1/3, 96-100 Skierniewice, Poland
* Correspondence: shaghef.ejaz@bzu.edu.pk (S.E.); walidmosa@alexu.edu.eg (W.F.A.M.)

**Abstract:** Climacteric rise in the rate of respiration and ethylene production in harvested persimmon (*Diospyros kaki*) fruits leads to early onset of fruit tissue senescence. Therefore, this study was carried out to maintain the quality of stored persimmons by using edible coatings. For this purpose, *Aloe vera* gel was combined with food hydrocolloids, gum arabic or tragacanth gum, and applied on persimmon fruits that were stored for 20 days at 20 ± 1 °C and 80–85% RH. Biocomposite coatings, compared to control, remarkably reduced weight loss, decay incidence, respiration rate, ethylene production, electrolyte leakage, malondialdehyde, $H_2O_2$ and superoxide anion content in stored fruits. The use of composite coatings inhibited colour change by reducing the accumulation of total carotenoids, maintained higher antioxidative enzymes activity and suppressed the activity of cell wall degrading enzymes, resultantly preserving cell wall components. Composite coated fruits exhibited the least change in biochemical attributes and higher sensory scores in comparison with non-coated fruits at the end of the storage period. In conclusion, *Aloe vera* gel/gum arabic was the most effective coating treatment before storing persimmons.

**Keywords:** persimmon; gum arabic; tragacanth gum; edible coating; mucilage; biocomposite





## 1. Introduction

The application of edible coatings or films minimizes quality deterioration and extend the postharvest life of fresh produces. Edible coatings or films may be classified as polysaccharides, proteins and lipids or their combinations [1]. Different polysaccharides such as chitosan, alginate, *Aloe vera* gel (AVG), gums and pectin are widely used to prolong the shelf/storage life of fresh produces [2]. AVG is a pulp-based gelatinous matrix derived from the leaf tissues of *Aloe vera* plant [3]. For preserving bioactive compounds and extending postharvest life, AVG has been applied on various fruits such as papaya [4], raspberry [5], guava [6] and persimmon [7].

Natural gums are hydrocolloids that are hydrated when added to water, resulting in gel formation or stabilizing emulsion systems [8]. Plant exudates such as gum arabic (GA) and tragacanth gum (TCG) are produced through a process known as gummosis caused by the decomposition of plant cellulose [9]. Application of GA preserves the antioxidants and quality attributes thereby reducing postharvest oxidative stress and limiting the activity of cell wall degrading enzyme in persimmon [10] and apricot fruits [11]. TCG has distinctive biological and chemical characteristics such as non-toxicity, oral safety, biocompatibility,

biodegradability and stability under a wide range of pH [12]. Ali et al. [13] reported that TCG preserved bioactive compounds, reduced activity of cell wall degrading enzymes, and thus extended shelf life of apricot fruits.

An edible coating may have more than one coating material that is applied using layer-by-layer technique [14,15] or as a single composite formulation [16–18]. Although, AVG possesses the antimicrobial and antioxidant properties, its insufficient film-forming ability confines its use on a wide scale. AVG might provide low barrier properties and allow water permeability to some extent [2]. To overcome this issue, film forming compounds such as starch, cellulose, gelatin, gellan gum etc. are incorporated in the AVG to prepare efficient biocomposite formulation [19,20]. Addition of AVG in a composite coating has shown improved bioactive compounds, antioxidative and antimicrobial properties [1]. Furthermore, composite films prepared with AVG and other compounds show improved barrier and physical properties. The structural and chemical interactions within AVG/gellan gum composite film enhanced its functional characteristics, resulting in improved water impenetrability in comparison with AVG alone. Additionally, the nano-mechanical characteristics in AVG/gellan gum composite film improved more than that of films based on AVG or gellan gum alone [19]. Reduced physiological weight loss and higher ascorbic acid content were found in AVG/guar gum coated mango fruits [17]. Application of AVG/GA limits the availability of $O_2$, thereby reducing the oxidation of organic acids in coated fruits [16]. Composite application of AVG and basil seed mucilage on apricot fruits considerably inhibited the physiological weight loss, total soluble solids, respiration rate and ethylene production, and retained higher titratable acidity, total phenolic content, total antioxidants and sensory quality [21]. Similarly, our previous work has shown that composite coating prepared from GA and CMC (carboxymethyl cellulose) considerably inhibited physiological weight loss, respiration rate, ethylene production and activity of cell wall degrading enzymes, and mitigated the postharvest oxidative stress by activating the antioxidant defence mechanism in tomatoes [18]. AVG in combination with TCG considerably lowers the physiological weight loss, colour change and tissue softening in stored mushrooms [22].

Persimmon (*Diospyros kaki*) fruits are a significant source of vitamins, minerals, dietary fibres, carbohydrates, carotenoids, condensed tannins, phenolic and antioxidants, which protect humans against oxidative stress-related diseases, and anti-mutagenic and anti-carcinogenic activities [23,24]. Persimmon is a climacteric fruit that deteriorates promptly when stored at room temperature; while during cold storage, the fruit is susceptible to chilling injury. Two of the most common chilling injury symptoms noticed in the fruit are external browning and softening [25]. Therefore, several scientists have reported the advantage of applying an edible coating based on sole coating material on persimmon fruits [7,10,26,27]. Nevertheless, to the best of our knowledge, no study has been reported on the application of a composite coating based on hydrocolloid gum and *Aloe vera* gel on the extension of postharvest life of persimmon fruits. Therefore, this experiment was conducted to extend the postharvest life of persimmon fruits by the application of composite coatings based on AVG, GA and TCG during ambient storage conditions.

## 2. Materials and Methods

### 2.1. Fruit Material

Persimmon fruits were harvested at the physiological maturity (yellowish-orange peel colour) stage from an orchard located in Swat, Pakistan in October 2022. The harvested fruits were immediately shifted (at $15 \pm 2$ °C, ~80% RH) to Postharvest Laboratory at Bahauddin Zakariya University, Multan, Pakistan. The fruits were sorted for uniformity in size, shape and colour (hue angle: 67.16 h°, chroma: 28.15) and were free from any visible injury or decay. The fruits were rinsed with distilled water and sanitized by immersing in a NaClO (0.01%) solution for 2 min. Subsequently, the fruits were air-dried at ~15 °C for 2 h.

## 2.2. Edible Coatings Prpeparation

Coating formulations were prepared according to our previously published work on AVG [7], GA [10] and TCG [28]. Briefly, *Aloe vera* leaves were harvested from 3-year-old plants. The outer cortex was separated from the parenchyma (gel matrix). The gel matrix was uniformly homogenized in a blender (MJJ-176P, Panasonic, Kadoma, Japan) at 12,000 rpm and subsequently filtered with a 2-layers of muslin cloth. Filtered *Aloe vera* gel was used to prepare different coating formulations. For preparing GA solution, GA powder (Daejung, Siheung-si, Republic of Korea) was added to distilled water followed by continuous stirring (1200 rpm at 40 °C) for 4 h. Aqueous solution of TCG was prepared by adding TCG powder (Sigma-Aldrich, St. Louis, MO, USA) in distilled water, followed by stirring (1200 rpm at 70 °C for 4 h) and blending in the blender (12,000 rpm for 15 min). Then the TCG solution was incubated at 4 °C for 24 h to hydrate the hydrocolloid gum [13].

AVG, AVG/GA and AVG/TCG formulations were prepared by mixing AVG solution with either distilled water, GA or TCG solution in a 1:1 ratio (*v*/*v*), respectively. The final concentrations of GA and TCG in their respective formulations were 10% and 1%, respectively. All formulations were stirred (1500 rpm) for complete homogenization at 40 °C for 2 h. Thereafter, 1.5% glycerine was added to all formulations for improving the wettability of the coatings.

## 2.3. Application of Edible Coatings and Storage

On the first day (D0) of the experiment, 15 fruits per replication (45 fruits/ treatments) were taken randomly from experimental fruits and immediately analysed for below given traits. Further, 900 persimmon fruits were equally distributed into 4 groups and treated with control (distilled water), 50% AVG, 50% AVG + 10% GA and 50% AVG + 1% TCG, respectively. Each treatment had 3 biological replications with 15 fruits per replication. Persimmon fruits were dipped in either distilled water or respective coating solutions for 3 min. Afterwards, all fruits were air-dried for 2 h at ~15 °C. The treated fruits were stored in clamshell clear polyethylene terephthalate (PET) boxes which had 5 mm holes on the top and sides. Finally, the packed fruits were stored for 20 days at 20 ± 1 °C and 80–85% RH. Different quality analyses were performed every 4th day during the whole experiment. For storing samples, liquid nitrogen was used to crush randomly selected fruits and homogenized samples were stored at −80 °C for further study.

## 2.4. Physiological Weight Loss (PWL)

The fresh fruits were weighed on a sampling day and PWL (%) was determined as a percent difference between D0 and the given sampling day.

## 2.5. Decay Incidence

Each fruit was visually examined and the fruits showing mycelia, rot or any disease spots ($\geq$1 mm diameter) were considered decayed. The number of decayed fruits over total number of fruits was presented as % decay incidence [18].

## 2.6. Respiration Rate and Ethylene Production

For the measurement of respiration rate and ethylene production, 2 persimmon fruits from each replication were placed in sealed air-tight plastic jars at ambient temperature for 1 h (25 °C). For this purpose, a gas analyzer (F-950, Felix Instruments, Camas, WA, USA) was used. The respiration rate was expressed in mmol $CO_2$ $kg^{-1}$ $h^{-1}$ and ethylene production in µmol $kg^{-1}$ $h^{-1}$.

## 2.7. Fruit Peel Colour

The fruit peel colour was examined by a colourimeter (CR-400, Konica Minolta, Tokyo, Japan). L*, a* and b* values were measured from 3 different areas of each fruit sample and used to calculate hue angle and chroma [29].

## 2.8. Total Carotenoid Contents

Total carotenoid contents were assessed as advised by Lichtenthaler [30]. 1 g of fruit sample was homogenized in acetone (80%) followed by centrifugation. Resulting residues were again homogenized and centrifuged. Both collected supernatants were combined. Afterwards, the absorbance was recorded at 470, 646 and 663 nm in a spectrophotometer (UV-1900, BMS, Toronto, ON, Canada) and carotenoid contents were denoted as $\mu g\ g^{-1}$ fresh weight (FW).

## 2.9. Electrolyte Leakage

15 discs, prepared from the fruit peel, were rinsed with deionized water, dipped in 50 mL of deionized water and shaken on an orbital shaker for 30 min. EC1 was noted by an EC meter (Mi-180 Bench Meter, Milwaukee Instruments, Szeged, Hungary). Later on, the mixture was boiled for 15 min and cool down before measuring EC2. Electrolyte leakage (%) was calculated by using the following formula: (EC1/EC2) $\times$ 100 [31].

## 2.10. Malondialdehyde (MDA) Content

15 mL of trichloroacetic acid (10%) was used for the homogenization of 1 g of fruit sample. The homogenized sample was centrifuged and 2 mL of 2-thiobarbituric acid (0.60%) was mixed with 2 mL supernatant. The reaction mixture was then heated in a water bath at 100 °C for 20 min. Upon cooling, absorbance was noted at 450, 532 and 600 nm wavelength in a spectrophotometer (UV-1900, BMS, Toronto, ON, Canada) and MDA concentration was denoted as $\mu mol\ kg^{-1}$ FW [32].

## 2.11. Hydrogen Peroxide ($H_2O_2$) Content

$H_2O_2$ concentration was estimated according to the protocol of Velikova and Loreto [33]. 1 g of fruit sample was homogenized with 0.1% (1 mL) of trichloroacetic acid and centrifuged at 12,000$\times$ $g$ (Silent spin CLP, Labnet, Edison, NJ, USA) for 15 min. Subsequently, the collected supernatant (500 μL) was mixed with 500 μL of phosphate buffer (PhB) (pH 7.0) and 1 mL of KI (1 M). After that, the absorbance was measured at 390 nm in a spectrophotometer (UV-1900, BMS, Toronto, ON, Canada) and the concentration of $H_2O_2$ was denoted as $\mu mol\ kg^{-1}$ FW.

## 2.12. Superoxide Anion Content

For the determination of superoxide anion content, 1 g of fruit sample was homogenized in 2 mL of PhB (50 mM, pH 7.8) and centrifuged at 10,000$\times$ $g$ (Silent spin CLP, Labnet International, Edison, NJ, USA) for 15 min. The content of superoxide anion radical was calculated by observing nitrite accumulation from hydroxylamine, and its concentration was denoted as $nmol\ min^{-1}\ kg^{-1}$ FW [31].

## 2.13. Enzymatic Antioxidants

2 mL of PhB (100 mM, pH 7.2) was used for grinding 1 g of fruit sample followed by centrifugation (Silent spin CLP, Labnet, Edison, NJ, USA) for the separation of supernatant [18]. The collected supernatant was used for the estimation of ascorbate peroxidase (APX), catalase (CAT), superoxide dismutase (SOD) and peroxidase (POD) activities.

For APX activity, a reaction mixture containing 100 μL of 0.5 mM L-ascorbate, 100 μL of 0.1 mM $H_2O_2$, 200 μL of 50 mM PhB (pH 5.0) and 100 μL of enzyme extract was prepared. The absorbance was monitored in a spectrophotometer (UV-1900, BMS, Toronto, ON, Canada) at 290 nm [34].

For the determination of CAT activity, the method of Ali et al. [34] was used. 100 μL of enzyme extract was added to a test tube containing 100 μL of 5.9 mM of $H_2O_2$ and 2.3 mL of 50 mM PhB (pH 5.0). Thereafter, the absorbance of the reaction mixture was noted at 240 nm in a spectrophotometer (UV-1900, BMS, Toronto, ON, Canada).

For the estimation of SOD activity, 100 μL of enzyme extract, 600 mL of deionized water, 500 μL of PhB (50 mM, pH 5.0), 250 μL of 0.15 μM Triton-X, 150 μL of 25 μM nitro

blue tetrazolium chloride, 150 μL of 0.65 μM riboflavin and 250 μL of 20 μM methionine were mixed in a test tube. The reaction mixture was exposed to UV light for 10 min and, thereafter, the absorbance was noted in a spectrophotometer (UV-1900 BMS, Toronto, ON, Canada) at 560 nm [34].

POD activity was estimated by using the method of Ali et al. [34]. In the test tube, the reaction mixture was prepared, containing 150 μL of 45 mM $H_2O_2$, 600 μL of PhB (50 mM, pH 5.0) and 150 μL of 25 mM guaiacol. For the initiation of the reaction, 100 μL of enzyme extract was added to the above prepared mixture, and absorbance was noted at 470 nm in a spectrophotometer (UV-1900, BMS, Toronto, ON, Canada).

Enzyme activities were stated as U $mg^{-1}$ protein. The protein content was estimated by following Bradford [35] assay and using bovine serum albumin as a standard.

### 2.14. Non-Enzymatic Antioxidants

### 2.14.1. Estimation of Antioxidant Activity

Ethanolic extract was obtained by grinding 1 g of fruit sample in 10 mL of ethanol (80%). The resulting homogenate was shaken and passed through filtration process. 1 mL of ethanolic extract was mixed with 2 mL of DPPH and then incubated in the dark conditions (25 °C) for 30 min. Later on, the absorbance was monitored at 520 nm in a spectrophotometer (UV-1900, BMS, Toronto, ON, Canada) and the antioxidant activity was stated in % [36].

### 2.14.2. Estimation of Total Phenolic Content (TPC) and Total Flavonoid Content (TFC)

In 10 mL of methanol (80%), 1 g fruit sample was grinded followed by centrifugation (Hermle Z-326 K, Labortechnik, Wasserburg, Germany) for the separation of methanolic extract. TPC and TFC were estimated in the collected supernatant according to the method used by Ebrahimi and Rastegar [17]. 300 μL of methanolic extract, 1.2 mL of $Na_2CO_3$ (7%) and 1.5 mL of Folin–Ciocalteu were mixed for the preparation of the reaction mixture which was incubated in dark conditions (25 °C) for 90 min. After that, the absorbance was recorded at 750 nm in a spectrophotometer (UV-1900, BMS, Toronto, ON, Canada). Galic acid was used for plotting the standard curve and TPC was stated as mg GAE 100 $g^{-1}$ FW.

Methanolic extract (500 μL) was mixed with $AlCl_3$ (100 μL, 10%) and acetate potassium (100 μL, 1 mM) solutions, and incubated (25 °C) for 30 min. Thereafter, the absorbance was monitored at 415 nm in a spectrophotometer (UV-1900, BMS, Toronto, ON, Canada). Quercetin was used for plotting the standard curve and TFC was denoted as mg QE 100 $g^{-1}$ FW.

### 2.14.3. Soluble Tannin

Soluble tannin was estimated by following the protocol of Taira [37] with minor modifications. In 10 mL of methanol (80%), 1 g of fruit sample was homogenised and subsequently centrifugated at 10,000× $g$ at 4 °C (Hermle Z-326 K, Labortechnik, Germany) for 10 min. The collected supernatant was mixed with Folin-Ciocalteu reagent, and after 5 min, $Na_2CO_3$ (10%) was added. After 1 h incubation (25 °C), the absorbance was monitored at 725 nm in a spectrophotometer (UV-1900, BMS, Toronto, ON, Canada). Soluble tannin was stated in %.

### 2.15. Cell Wall Components

The methods of Wang et al. [38] were adopted for the estimation of protopectin, water-soluble pectin (WSP), cellulose and hemicellulose content in stored persimmon fruits. 1 g of composite fruit sample was boiled in 25 mL of ethanol (95% *v/v*) for 30 min. After cooling, the homogenate was centrifuged for 15 min at 8000× $g$ and 4 °C. The collected sediment was dissolved in 25 mL of ethanol (95%, *v/v*), then boiled and centrifuged according to the procedure described above. The same process was repeated thrice. After that, the residue was homogenized in 20 mL of deionized water and heated at 50 °C in a water bath for 30 min. The resultant homogenate was centrifuged at 8000× $g$ for 15 min. The collected supernatant was used for the estimation of WSP. The remaining sediment was dissolved in 25 mL of $H_2SO_4$ (0.5 M) and boiled for 1 h. The resultant homogenate was centrifuged at

8000× $g$ for 15 min, and the supernatant was used for the estimation of protopectin. The protopectin and WSP content were denoted as g kg$^{-1}$ FW.

30 g of frozen fruit pulp was ground with 200 mL of ethanol (80%, $v/v$). The homogenate was boiled for 1 h and filtered. The residue was washed thrice by using ethanol (80%) and then soaked in $C_2H_6OS$ (90% $v/v$) for 10 h. Filtration was repeated, and the residue was washed thrice with 100 mL of acetone. Afterwards, vacuum drying (40 °C) was performed and the residue was designated as cell wall material.

300 mg of cell wall material was homogenized in 10 mL of acetic acid buffer (50 mM, 6.5 pH) and centrifuged at 10,000× $g$ at 4 °C for 10 min. The sediment was collected after decanting of the supernatant and dissolved in 10 mL of acetate buffer (50 mM, 6.5 pH) which consisted of cyclohexane-trans-l, 2-diamine tetraacetate (2 mM). Subsequently, the mixture was stirred continuously for 6 h and centrifuged at 10,000× $g$ at 4 °C for 10 min. The lower precipitate was dissolved in $Na_2CO_3$ (50 mM) which contained cyclohexane-trans-l, 2-diamine tetraacetate (2 mM) and again stirred for 6 h before centrifuging at 10,000× $g$ at 4 °C for 10 min. The residue was dissolved in 10 mL of 4 NaOH (mM) solution that contained $NaBH_4$ (100 mM) and centrifuged for 10 min at 10,000× $g$. The collected sediment was designated as cellulose materials and used for the estimation of cellulose and hemicellulose content. The cellulose and hemicellulose content were denoted as mg kg$^{-1}$ FW.

*2.16. Fruit Firmness and Activity of Cell Wall Degrading Enzymes*

A digital handheld fruit penetrometer equipped with 8 mm tip (FR-5120, Lutron Electronics Enterprises, Taiwan, China) was used for the measurement of fruit firmness (Newton: 'N') of 5 fruits per replication.

Pectinmethylesterase (PME) activity was measured with the modified method used by Saleem et al. [10]. Shortly, enzyme extract was collected by using a NaCl (8.8%) solution followed by adjusting the pH of the enzyme extract with sodium hydroxide solution. A reaction mixture was prepared by adding 750 µL of deionized water, 150 µL of bromothymol blue (0.01%, pH 7.5) and 2 mL of pectin. The reaction was initiated by adding 100 µL of enzyme extract and observing the enzyme activity at 620 nm in a spectrophotometer (UV-1900, BMS, Toronto, ON, Canada). PME activity was stated as U mg$^{-1}$ protein.

Extraction for polygalacturonase (PG) and cellulase (CEL) were carried out according to Saleem et al. [10]. Briefly, pre-cooled sodium acetate buffer (8 mL) along with NaCl (2 mL) was used for grinding fruit sample (1 g), followed by incubation and centrifugation.

PG activity was estimated by following the protocol of Yoshida et al. [39] with slight modifications. 400 µL of pectin (1%), 100 µL of crude extract and 500 µL of sodium acetate buffer (0.2 M, pH 4.5) were added to a test tube and incubated at 37 °C for 1 h. After incubation, 1 mL of dinitrosalicylic acid was added to the mixture followed by boiling for 5 min and then cooling at ambient temperature. The activity was observed at 540 nm in a spectrophotometer (UV-1900, BMS, Toronto, ON, Canada) and stated as U mg$^{-1}$ protein.

For CEL activity, 400 µL of 1% carboxy methylcellulose, 100 µL of crude extract and 500 µL of sodium acetate buffer (0.1 mM, pH 5.0) were mixed in a test tube and incubated at 37 °C for 1 h. Afterwards, 1 mL of 1% dinitrosalicylic acid was added to the resulting mixture and boiled for 5 min. After cooling the reaction mixture, the activity was observed at 540 nm in a spectrophotometer (UV-1900, BMS, Toronto, ON, Canada) and stated as U mg$^{-1}$ protein [40].

*2.17. Biochemical Analysis*

2.17.1. Total Soluble Solids (TSS)

Fruit juice was extracted through homogenization of persimmon fruits in an electric blender followed by filtration through a double layered muslin cloth. TSS was measured with the help of digital refractometer (Atago PAL-1, Atago Co., Tokyo, Japan) and expressed in %.

### 2.17.2. Titratable Acidity (TA)

Extracted fruit juice (10 mL) was diluted up to 50 mL by adding distilled water and, subsequently, titrated against NaOH (0.10 N) up to pH 8.1 (Mi-180 Bench Meter, Hungary) to quantify percent (%) titratable acids [21].

### 2.17.3. Ascorbic Acid

For the estimation of ascorbic acid concentration, 10 mL of fruit juice was diluted up to 100 mL with 0.4% oxalic acid before filtering it. The resulting aliquot (10 mL) was titrated against 2, 6 dichloroindophenol until the development of light pink colour [34].

### 2.18. Sensory Evaluation

Various sensory aspects (taste, aroma and overall acceptability) were judged on a 9-point hedonic scale by a trained panel [13]. The scale values ranged from 9 (like extremely) to 1 (dislike extremely).

### 2.19. Statistical Analysis

The data were analysed statistically by using statistical software Statistix version 8.1 analytical software (Tallahassee, FL, USA) to evaluate the effects of edible coatings on the postharvest life of persimmon fruits under storage conditions. The treatment means were compared by using Fisher's Least Significant Difference test ($p \leq 0.05$).

## 3. Results and Discussion

### 3.1. Physiological Weight Loss (PWL)

A steady increase in PWL was observed as the storage days increased (Figure 1A). However, the fruits coated with biocomposite formulations had significantly less PWL compared to AVG coated and non-coated fruits. Among biocomposite coatings, AVG/GA significantly lowered PWL on all sampling days. On D20, AVG/GA coated persimmons exhibited the lowest PWL (9.97%) as compared to AVG/TCG (11.17%), AVG (12.56%) and non-coated fruits (15.23%). PWL determines the postharvest quality and life of fruits and vegetables. Fresh fruits and vegetables lose mass through respiration and transpiration; the latter is affected by the vapour pressure gradient between the fruit tissue and the surroundings [3,18,41]. AVG-based edible coatings provide a physical barrier, and hence slow down respiration (by limiting $O_2$ supply) and transpiration (by reducing water loss) processes in coated fruits [3,21]. Our results are similar to previous findings, where reduced PWL has been observed in guava and mushrooms coated with an AVG-based biocomposite [16,22].

### 3.2. Decay Incidence

Up to D4 of storage, there was no decay incidence in any treatment (Figure 1B). However, on D8, non-coated and AVG coated fruits showed decay incidence (8 and 1%, respectively). From D12 onward, although all treatments showed decay incidence, but biocomposite coated fruits had the minimum decay incidence. On D20, non-coated persimmons had the highest decay incidence that was 33% higher than the AVG/GA coated fruits and 27% higher than AVG/TCG coated fruits. Edible coatings suppress the growth of some fungal diseases and prevent the disease-related losses in fruits [1,16]. Plant-based gum application functions as a barrier between disease causing microorganisms and fruits, resulting in a lower disease incidence [11,16]. Suppressed decay incidence has been documented in AVG-based coated fruits such as raspberry [5], papaya [4] and sapodilla [3].

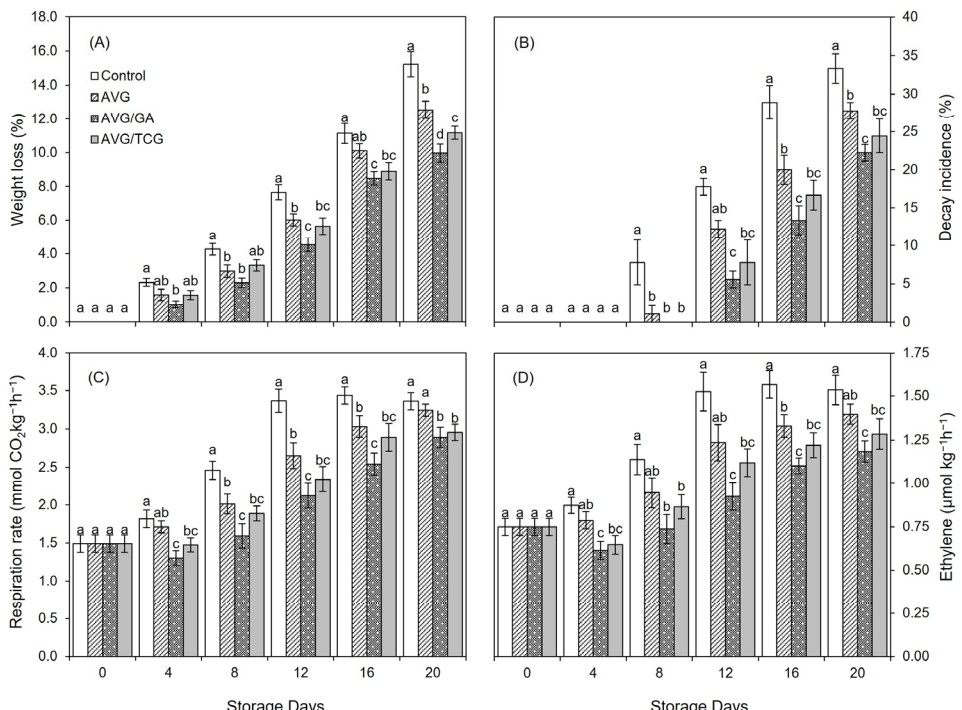

**Figure 1.** Effect of AVG-based composite coatings on physiological weight loss (**A**), decay incidence (**B**), respiration rate (**C**) and ethylene production (**D**) of persimmon fruits during 20 days storage (analysis on day 0, 4, 8, 12, 16 and 20) at $20 \pm 1$ °C and 80–85% relative humidity. All standard error bars are centered on means (3 biological replicates). Graph bars sharing same letters on a storage interval have non-significance difference according to the LSD test at $p \leq 0.05$.

### 3.3. Respiration Rate and Ethylene Production

During the first 4 days of storage, the fruits coated with biocomposite, especially AVG/GA, exhibited lower respiration rate as compared to the non-coated fruits (Figure 1C). Non-coated fruits showed an early respiration peak on D12. However, the application of coatings caused significant inhibition in respiration rate up to D20. At the end of the experiment, non-coated persimmons showed higher respiration rate (3.36 mmol $CO_2$ kg$^{-1}$ h$^{-1}$), which was 1.16- and 1.14-fold higher than AVG/GA and AVG/TCG treated persimmons, respectively. Composite coated persimmons showed lower ethylene production as compared to AVG coated and non-coated fruits in the first 4 days of storage (Figure 1D). Except control, all treatments showed a climacteric rise in ethylene production between D8 and D20. On D20, AVG/GA coated fruits showed the least ethylene production (1.18 µmol kg$^{-1}$ h$^{-1}$), and non-coated fruits the maximum (1.53 µmol kg$^{-1}$ h$^{-1}$).

In this study, reduced respiration rate and ethylene production resulted in delayed ripening and senescence of gum coated persimmon fruits, as have been reported in sweet cherry and tomato fruits [18,42]. Coatings behave as a barrier and limit the $O_2$ and $CO_2$ permeability through the fruit surface, thereby modifying internal atmosphere physiology [21,41]. Reduced respiration and ethylene production have been reported in the fruits coated with plant-based edible coatings; for example, in sweet cherry [42], apricot [21] and tomato [18].

### 3.4. Fruit Peel Colour (Hue Angle and Chroma)

Hue angle decreased with storage time in all treatments (Figure 2A). From D8, a significant difference between treatments was observed. On D20, AVG/GA coated persimmon fruits showed the highest hue angle (46.24 h°) and non-coated fruits had the lowest hue angle (39.00 h°). Hue angle reduces as ripening progresses in persimmons and colour transforms from yellow-orange to red-orange [43]. The accumulation of carotenoids also led to a decline in hue angle [44]. The greater hue angle in coated persimmons could be

attributed to the ability of edible coatings to suppress respiration rate, ethylene production (Figure 1C,D), carotenoid accumulation and senescence by modifying atmosphere around the fruit's surface [18,45]. Our findings are similarly to previous investigations, where edible coating application considerably preserved a greater hue angle in coated fruits including fresh-cut persimmon [26], mango [45] and tomato fruits [18].

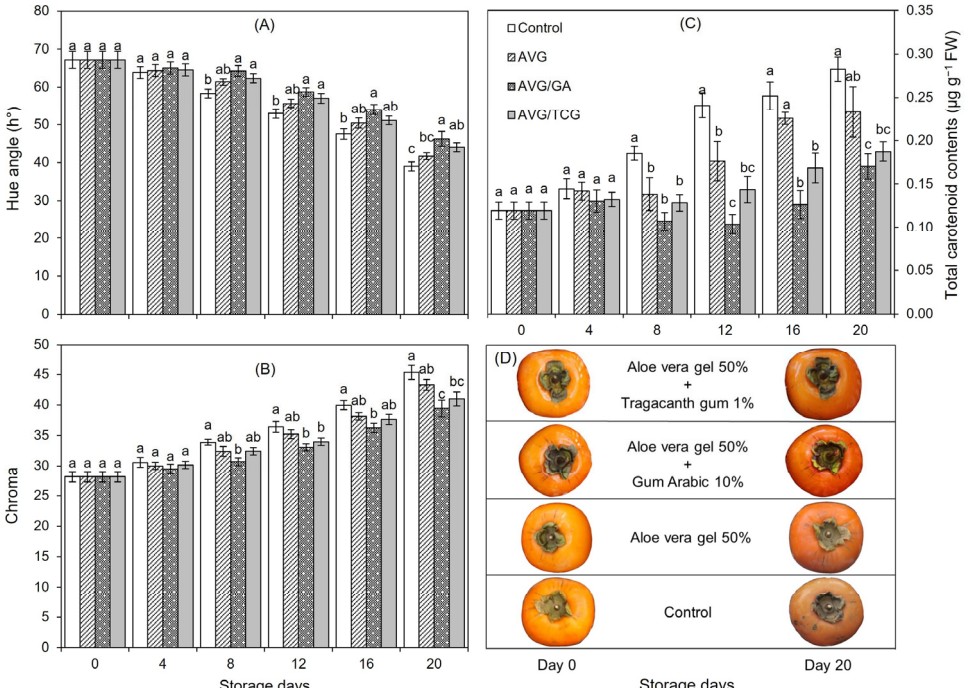

**Figure 2.** Effect of AVG-based composite coatings on hue angle (**A**), chroma (**B**) total carotenoid contents (**C**) and appearance (**D**) of persimmon fruits during 20 days storage (analysis on day 0, 4, 8, 12, 16 and 20) at 20 ± 1 °C and 80–85% relative humidity. All standard error bars are centered on means (3 biological replicates). Graph bars sharing same letters on a storage interval have non-significance difference according to the LSD test at $p \leq 0.05$.

Chroma increased with the development of storage days (Figure 2B). However, composite coatings application, especially AVG/GA, considerably inhibited the development of chroma in persimmon fruits. On D20, non-coated and AVG coated persimmon fruits showed higher chroma value (45.33 and 43.28, respectively) as compared to AVG/GA and AVG/TCG coated fruits (39.43 and 40.98, respectively).

Chroma is an indicator of colour saturation; a higher chroma value indicates higher saturation of colours [45]. It has been observed that chroma increases with the ripening process due to colour development in persimmon fruits [43]. The lower chroma values in composite coated persimmon fruits was most likely due to the effect of edible coating, which slowed down ripening and senescence by reducing the respiration rate, ethylene production and carotenoid accumulation [45].

### 3.5. Total Carotenoid Contents

Total carotenoid contents were significantly affected by the treatments in stored persimmon fruits (Figure 2C). With the advancement of storage days, non-coated persimmon fruits showed a steady increase in total carotenoid contents (0.119 to 0.283 µg g$^{-1}$ FW) as compared to the coated fruits. Additionally, On D20, lower carotenoid contents (0.170 and 0.187 µg g$^{-1}$ FW) were exhibited by AVG/GA and AVG/TCG coated persimmon fruits, respectively, as compared to the AVG coated and non-coated fruits (0.283 and 0.233 µg g$^{-1}$ FW, respectively).

Persimmon fruits contain a high concentration of carotenoids at the full ripening stage when fruit colour changes from yellow-orange to red-orange [25]. Application of edible coating postpones the accumulation of carotenoids by regulating the fruit ripening in stored persimmon [10] and apricot fruits [21]. Higher hue angle and lower chroma value (Figure 2A,B) in AVG-based composite coated persimmon fruits might be due to reduced accumulation of total carotenoid contents in coated fruits. Therefore, the fruits coated with composite coating, especially with AVG/GA, had marketable appearance on D20 (Figure 2D).

### 3.6. Electrolyte Leakage

Electrolyte leakage was considerably influenced by the storage days, as gradual progression was noticed in all treatments from D0 to D20 (Figure 3A). However, the use of composite coatings greatly delayed the increment in electrolyte leakage from D4 to D20 as compared to AVG or control. On D20, AVG/GA coated persimmon fruits showed the least electrolyte leakage (45.57%) and non-coated fruits the most (61.64%).

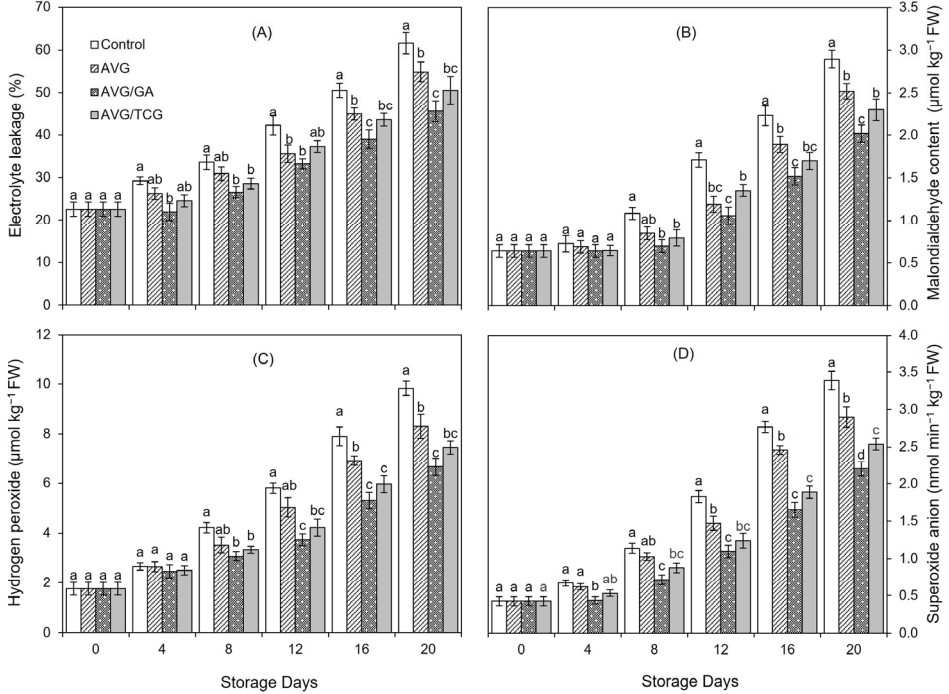

**Figure 3.** Effect of AVG-based composite coatings on electrolyte leakage (**A**), malondialdehyde content (**B**) hydrogen peroxide (**C**) and superoxide anion content (**D**) of persimmon fruits during 20 days storage (analysis on day 0, 4, 8, 12, 16 and 20) at 20 ± 1 °C and 80–85% relative humidity. All standard error bars are centered on means (3 biological replicates). Graph bars sharing same letters on a storage interval have non-significance difference according to the LSD test at $p \leq 0.05$.

Climacteric fruits generally show a gradual increment in electrolyte leakage with progressed storage days [46]. Production of reactive oxygen species (ROS) cause oxidative damage that leads to the disintegration of the cell membrane and as a result increases the electrolyte leakage in stored fruits [11]. Reduced electrolyte leakage in coated persimmon fruits might be due to the effect of edible coating, as plant-based edible coatings maintain the integrity of the cell membrane thereby alleviating the postharvest oxidative stress during storage conditions [46]. Reduced electrolyte leakage has been observed in persimmon [10], apricot [11] and guava fruits [46] coated with natural gums.

### 3.7. Malondialdehyde (MDA) Content

A continuous advancement was observed in MDA concentration from D0 to D20 (Figure 3B). The use of edible coatings played an inhibitory role against MDA production in coated persimmon fruits throughout the storage days. On D20, AVG/GA coated persimmons exhibited the lowest MDA content (2.02 $\mu$mol kg$^{-1}$), whereas non-coated fruits had the highest MDA level (2.90 $\mu$mol kg$^{-1}$).

As an end product of lipid peroxidation, MDA usually increases during postharvest storage [7], and also the structure of the cellular membranes is disrupted [46]. Excessive generation of ROS, such as $H_2O_2$, is closely associated with increased oxidative stress and enhanced MDA production in harvested fruits [11]. The application of edible gums suppresses the accumulation of ROS by increasing the accumulation of antioxidants thereby reducing the production of MDA in coated fruits, such as apricot [11,13], persimmon [10], guava [46] and sweet cherry [42].

### 3.8. Hydrogen Peroxide ($H_2O_2$) Content

$H_2O_2$ production showed a gradual rise (1.75 to 8.07 $\mu$mol kg$^{-1}$ FW) with the advancement of storage days (Figure 3C). Up to D4 of storage, no significant difference was observed among the treatments. From D16, a sharp progression was seen in $H_2O_2$ production which continued up to the end of the experiment; however, composite coated fruits showed a lower $H_2O_2$ production as compared to AVG coated and non-coated fruits. Moreover, on D20, non-coated persimmon fruits exhibited 47% higher $H_2O_2$ content than AVG/GA coated fruits.

Postharvest storage induces oxidative stress which triggers the generation of $H_2O_2$ in harvested persimmons [10]. The application of plant-based gums as an edible coating decreases the production of free radicals by lowering postharvest oxidative stress [13]. Our results were in accordance with prior findings where the use of plant-based edible coatings reduced oxidative stress and suppressed the accumulation of $H_2O_2$ in apricot [11] and tomato fruits [18].

### 3.9. Superoxide Anion Content

Superoxide anion content increased in all treatments with storage time (Figure 3D). However, composite coated fruits exhibited a significantly reduced accumulation of superoxide anion. On D20, non-coated and AVG coated fruits had 1.53-folds and 1.31-folds higher superoxide anion, respectively, than AVG/GA coated persimmons.

Excessive ROS production is injurious to fresh produces during postharvest storage. A proper balance is essential between ROS production and its scavenging mechanism [47]. Edible coating application lowers the production of ROS, such as $H_2O_2$ and superoxide anion, in coated fruits [48]. The lower production of superoxide anion in composite coated fruits might be due to the higher enzymatic (Figure 4A–D) and non-enzymatic antioxidants (Figure 5A–D) in composite coated fruits. Suppressed accumulation of superoxide anion content has been reported in coated fruits, such as litchi and mango fruits [48,49].

### 3.10. Antioxidant Enzyme Activities

Increase in storage days caused a continuous reduction in APX activity across all treatments. APX activity slightly improved on D8 and again continued to reduce until the end of storage days (Figure 4A). However, the application of edible coatings significantly maintained higher APX activity in coated persimmon fruits. On D20, AVG/GA coated persimmon fruits exhibited the highest APX activity (186.78 U mg$^{-1}$ protein), followed by AVG/TCG coated fruits (170.79 U mg$^{-1}$ protein), whereas non-coated persimmon fruits showed the minimum APX activity (148.45 U mg$^{-1}$ protein).

CAT activity decreased (22.13 to 11.11 U mg$^{-1}$ protein) steadily with the passage of storage time regardless of the treatments (Figure 4B). On D20, a sharp decline in CAT activity was noticed in AVG coated and non-coated persimmons, while composite coated fruits

maintained a higher CAT activity. On D20, non-coated persimmon fruits had 17%, 35% and 27% lower CAT activity than AVG, AVG/GA and AVG/TCG coated fruits, respectively.

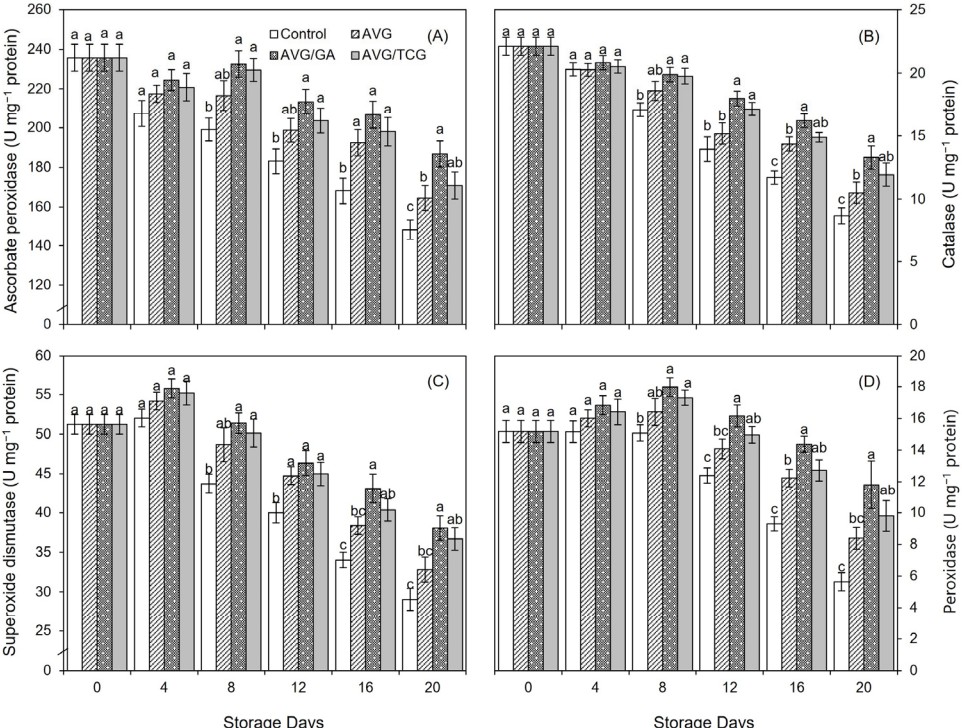

**Figure 4.** Effect of AVG-based composite coatings on ascorbate peroxidase activity (**A**), catalase activity (**B**), superoxide dismutase activity (**C**) and peroxidase activity (**D**) of persimmon fruits during 20 days storage (analysis on day 0, 4, 8, 12, 16 and 20) at 20 ± 1 °C and 80–85% relative humidity. All standard error bars are centered on means (3 biological replicates). Graph bars sharing same letters on a storage interval have non-significance difference according to the LSD test at $p \leq 0.05$.

Initially, up to D4, SOD activity increased in all treatments, but SOD activity was much higher in composite coated persimmon fruits (Figure 4C). From D8, a declining trend was noted in all treatments which continued till the end of storage days. On D20, AVG/TCG and AVG/GA coated persimmon fruits showed a higher SOD activity (38.05 and 36.68 U mg$^{-1}$ protein, respectively) followed by AVG coated fruits (33.84 U mg$^{-1}$ protein), whereas non-coated fruits exhibited the least SOD activity (29.03 U mg$^{-1}$ protein).

Up to D8 of storage, POD activity increased in coated persimmons, whereas it continuously decreased in non-coated fruits (Figure 4D). From D12 onward, all treatments exhibited a steady decline in POD activity. On D20, AVG/GA and AVG/TCG coated fruits exhibited 2.09-fold and 1.74-fold higher POD activity, respectively, than non-coated fruits.

During storage conditions, persimmon fruits are subjected to oxidative stress and thus produce more ROS [10]. APX, CAT, SOD and POD are the main enzymatic antioxidants for mitigating the postharvest oxidative stress in harvested fruits [11,50]. The use of plant-based gums as edible coatings may activate the antioxidant defence system which combats ROS and delays senescence in stored fruits [10]. AVG-based coating application maintains higher activity of antioxidative enzymes and protects fruit tissues from ROS, thus extending their postharvest life during storage conditions [6]. Higher activity of the antioxidant enzymes in composite coated persimmons may be due to the synergistic effect of AVG and gums, as individual application of AVG or plant-based gums has been shown to retain higher antioxidant enzymes activity in various fruits such as guava, persimmon and raspberry [5–7,10].

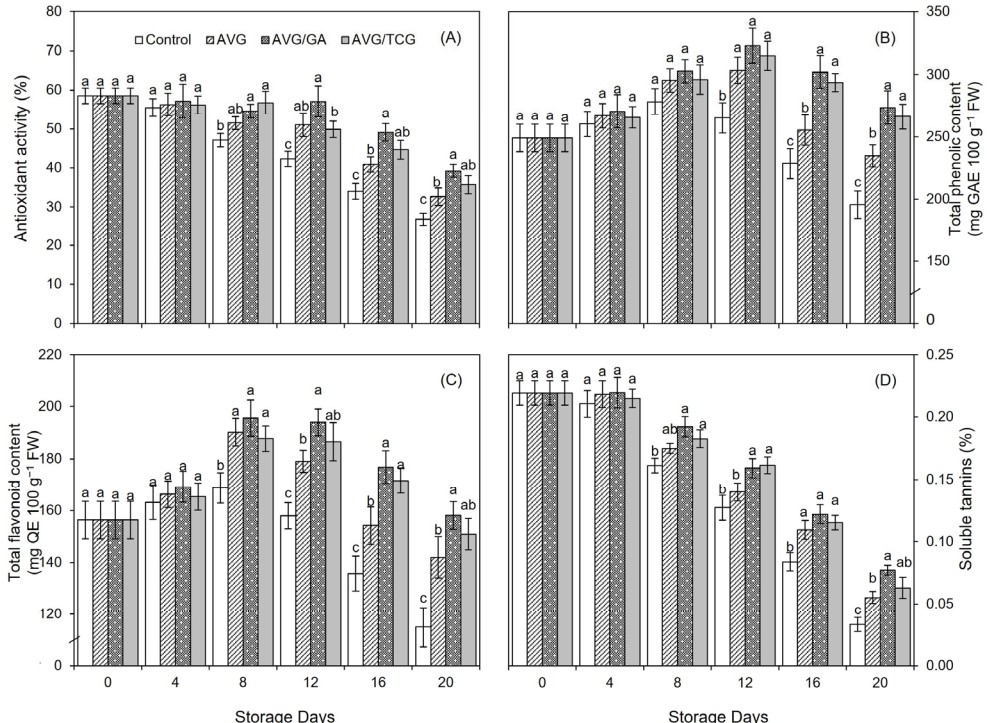

**Figure 5.** Effect of AVG-based composite coatings on antioxidant activity (**A**), total phenolic content (**B**), total flavonoid content (**C**) and soluble tannins (**D**) of persimmon fruits during 20 days storage (analysis on day 0, 4, 8, 12, 16 and 20) at $20 \pm 1$ °C and 80–85% relative humidity. All standard error bars are centered on means (3 biological replicates). Graph bars sharing same letters on a storage interval have non-significance difference according to the LSD test at $p \leq 0.05$.

### 3.11. Non-Enzymatic Antioxidants

#### 3.11.1. Antioxidant Activity

The antioxidant activity of persimmon fruits decreased with the storage period (Figure 5A).The reduction in antioxidant activity was maximum in non-coated and AVG coated persimmons. Moreover, on D20, AVG/GA coated persimmons exhibited 1.46- and 1.20-fold higher antioxidant activity than non-coated and AVG coated fruits, respectively.

Non-enzymatic antioxidants (such as ascorbic acid, carotenoids and phenolics) contribute to the antioxidant activity [10]. Their concentration decreases with the passage of storage time due to ageing in harvested fruits [18]. AVG is enriched with a large number of antioxidants which may contribute to maintaining the higher antioxidants in coated fruits [3,5]. Likewise, the use of edible gums also preserves higher antioxidants in coated apricot [11,13] and mango fruits [17]. The higher antioxidant activity in AVG/GA coated persimmon fruits might be due to the composite effect of AVG and GA.

#### 3.11.2. Total Phenolic Content (TPC) and Total Flavonoid Content (TFC)

The coated persimmon fruits showed a gradual increment in TPC up to D12, whereas TPC decreased in non-coated fruits after D8 (Figure 5B). From D16 to D20, all treatments exhibited a gradual reduction in TPC. During this period, composite coatings greatly inhibited the degradation of TPC. On D20, AVG/GA and AVG/TCG coated persimmon fruits showed higher TPC (273.75 and 266.41 mg GAE 100 g$^{-1}$ FW, respectively) in comparison with the non-coated fruits (195.59 mg GAE 100 g$^{-1}$ FW).

TFC initially improved up to D8 and then continued to decline across all treatments till the end of the storage period (Figure 5C). Nevertheless, the application of composite coatings conserved TFC in coated persimmon fruits. On D20, AVG/GA coated persimmon fruits exhibited higher TFC (158.09 mg QE 100 g$^{-1}$ FW), followed by AVG/TCG (150.84 mg

QE 100 g$^{-1}$ FW) and AVG coated fruits (141.97 mg QE 100 g$^{-1}$ FW), whereas non-coated fruits showed the least TFC (114.91 mg QE 100 g$^{-1}$ FW) on that day.

Phenolic compounds are important secondary metabolites that had significant antioxidant potential for scavenging ROS thereby regulating the functions of different enzymes [17]. Edible coatings play an important role in TPC retention because they modify the fruit's physiology by limiting respiration and oxidation and thus reduce the production of polyphenol oxidase [42,51]. Similarly, preserved TPC has been reported in natural gum-coated mango [17], guava [16] and apricot fruits [11,13].

Consistent with the findings of Anjum et al. (2020), GA-based coating conserved higher TFC in coated guava fruits. Likewise, higher TFC was also stated by Mendy et al. [4] and Khaliq et al. [3] while using AVG-based edible coating on papaya and sapodilla fruits, respectively. So, higher TFC in composite coated persimmon fruits might be due to the composite effect of AVG and gums.

### 3.11.3. Soluble Tannins

Soluble tannins exhibited a steady reduction in both coated and non-coated persimmon fruits (Figure 5D); however, this reduction was higher in non-coated fruits in comparison with composite coated fruits. On D20, AVG/GA coated persimmon fruits showed higher soluble tannin (2.32-fold) as compared to the AVG coated and non-coated fruits.

Reduction in soluble tannin is mainly associated with the ripening and senescence processes and is also influenced by the storage duration [52]. Similarly, fruit softening also cause the reduction of soluble tannin during post-harvest stage [28,37]. Higher soluble tannin in composite coated fruits may be due to the synergistic effect of AVG and edible gums [28].

### *3.12. Cell Wall Components*

The protopectin content decreased with storage days (Figure 6A). The maximum reduction in protopectin was recorded in non-coated persimmon fruits, followed by AVG coated fruits. On D20, AVG/GA coated persimmon fruits showed 1.26-fold higher protopectin content than non-coated fruits, which showed the least protopectin content (4.81 g kg$^{-1}$ FW) on that day.

Water-soluble pectin (WSP) showed an increasing trend with increasing storage days (Figure 6B). However, the use of composite coatings significantly reduced the content of WSP in coated persimmon fruits. On D20, persimmon fruits coated with AVG/GA showed a significantly lower content of WSP (7.27 g kg$^{-1}$ FW) than the non-coated fruits (8.63 g kg$^{-1}$ FW).

The cellulose content declined from D0 to D20 in persimmon fruits. However, the use of composite coatings significantly inhibited the reduction of cellulose content in coated fruits (Figure 6C). Likewise, On D20, AVG/GA and AVG/TCG coated persimmon fruits showed higher cellulose content (79.93 and 73.86 mg kg$^{-1}$ FW, respectively) as compared to AVG coated and non-coated fruits (72.28 and 63.47 mg kg$^{-1}$ FW, respectively). All treated persimmon fruits had reduced hemicellulose content with storage time (Figure 6D). On D20, AVG/GA coating preserved a higher hemicellulose content (44.00 mg kg$^{-1}$ FW), followed by AVG/TCG (40.65 mg kg$^{-1}$ FW) and AVG (34.09 mg kg$^{-1}$ FW). Non-coated fruits had the minimum hemicellulose content (25.08 mg kg$^{-1}$ FW).

The major component of the cell wall is pectin, which contributes to intercellular adhesion and strengthens the plant cell wall. Protopectin is degraded into WSP during fruit softening [27,38]. Likewise, fruit softening is also influenced by the cellulose and hemicellulose content, which normally degrade during fruit ripening. The application of edible coating maintained a lower amount of WSP, and a higher amount of protopectin, cellulose and hemicellulose content in persimmon fruit tissues, which effectively delayed softening of stored persimmon fruits. Conclusively, composite coatings postponed the senescence by minimizing the activity of cell wall degrading enzymes (Figure 7B–D) and stabilizing cell wall components. Our results are also confirmed by Xue et al. [27], who found that

application of composite (carboxymethyl-chitosan) coating delayed the degradation of cell wall components thereby maintaining greater firmness in coated persimmon fruits.

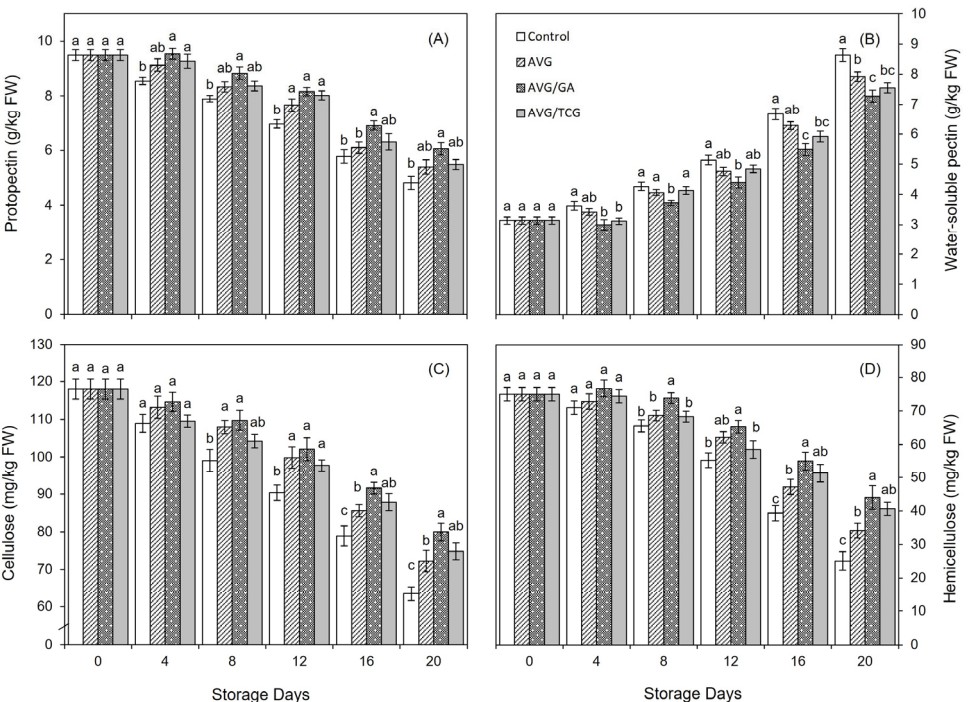

**Figure 6.** Effect of AVG-based composite coatings on protopectin content (**A**), water-soluble pectin content (**B**), cellulose content (**C**) and hemicellulose content (**D**) of persimmon fruits during 20 days storage (analysis on day 0, 4, 8, 12, 16 and 20) at 20 ± 1 °C and 80–85% relative humidity. All standard error bars are centered on means (3 biological replicates). Graph bars sharing same letters on a storage interval have non-significance difference according to the LSD test at $p \leq 0.05$.

### 3.13. Fruit Firmness and Activity of Cell Wall Degrading Enzymes

Fruit firmness reduced with the passage of time in all treatments (Figure 7A). However, the use of composite coatings significantly inhibited the loss of firmness. Moreover, on D20, non-coated fruits exhibited the minimum firmness (13.77 N), which was 70% lower than the firmness of AVG/GA coated fruits and 61% lower than the firmness of AVG/TCG coated fruits.

For PME activity, no significant effect was observed among the applied treatments up to D4 of storage (Figure 7B). On D20, non-coated persimmon fruits exhibited significantly higher PME activity (29.20 U mg$^{-1}$ protein), followed by AVG coated fruits (25.32 U mg$^{-1}$ protein). However, AVG/GA coated fruits exhibited the minimum PME activity (19.07 U mg$^{-1}$ protein). The application of composite coatings substantially suppressed PG activity (Figure 7C). On D20, non-coated and AVG coated persimmon fruits exhibited higher PG activity (21.86 and 18.66 U mg$^{-1}$ protein, respectively) in comparison with composite coated fruits. Irrespective of treatments, CEL activity increased with storage days (Figure 7D). On D20, compared to non-coated and AVG coated fruits, AVG/GA coated fruits showed 42% and 21% lower CEL activity, respectively.

Disintegration of cell wall polysaccharides occurs during fruit ripening and senescence which causes the loss of fruit firmness. Cell wall degrading enzymes such as PG, PME and CEL are responsible for this softening of fruits [23]. The use of plant-based gums as edible coatings maintains higher fruit firmness and inhibits the activities of hydrolases, such as PG, PME and CEL, which ultimately postpone softening and prolong the shelf life of harvested fruits [13]. Moreover, loss in fruit firmness is also linked to higher respiration rate and ethylene activity [44]. Supressed activity of cell wall degrading enzymes in AVG-based composite coated persimmon fruits might be due to reduced respiration rate and ethylene

production (Figure 1C,D) which otherwise cause fruit senescence and softening [18]. More-over, composite effect of polysaccharides-based coatings may have resulted in higher cell integrity due to reduced electrolyte leakage and MDA accumulation [10,13].

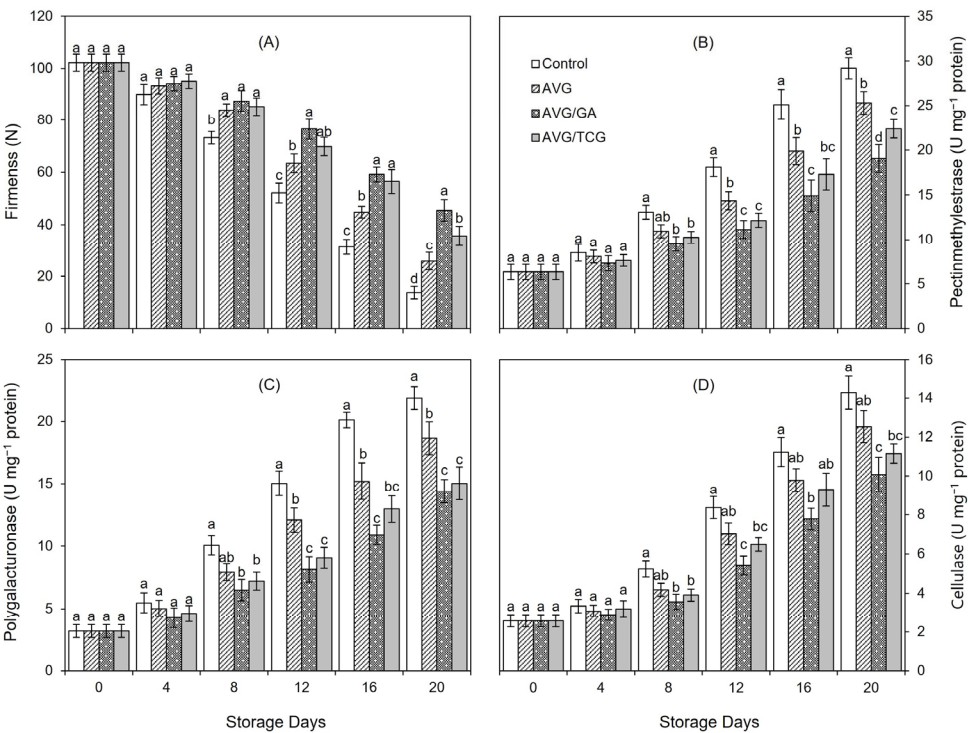

**Figure 7.** Effect of AVG-based composite coatings on firmness (**A**), pectinmethylesterase activity (**B**), polygalacturonase activity (**C**) and cellulase activity (**D**) of persimmon fruits during 20 days storage (analysis on day 0, 4, 8, 12, 16 and 20) at $20 \pm 1$ °C and 80–85% relative humidity. All standard error bars are centered on means (3 biological replicates). Graph bars sharing same letters on a storage interval have non-significance difference according to the LSD test at $p \le 0.05$.

### 3.14. Biochemical Analysis

3.14.1. Total Soluble Solids (TSS)

TSS increased from 12.95% on D0 to $20.35 \pm 1$% on D20 in all treated fruits (Table 1). Up to D4 of storage, no significant effect was observed among the treatments. From D12, the effect of applied coatings was noticed, where composite coated fruits showed lower and statistically similar TSS values. The highest TSS (21.35%) was found in the non-coated persimmon fruits at the end of the storage period, whereas the lowest level of TSS (19.65%) was noted in fruits coated with AVG/GA.

**Table 1.** Effect of *Aloe vera* gel-based composite coatings on total soluble solids (TSS), titratable acidity (TA), ascorbic acid, taste, aroma and overall acceptability in persimmon fruits during 20 days storage (20 ± 1 °C and 80–85% RH).

| Storage Days | Treatments | TSS (%) | TA (%) | Ascorbic Acid (mg 100 g$^{-1}$ FW) | Taste (Score) | Aroma (Score) | Overall Acceptability (Score) |
|---|---|---|---|---|---|---|---|
| D0 | | 12.95 | 0.553 | 56.62 | 5.17 | 5.42 | 5.17 |
| D4 | Control | 14.40 a | 0.516 a | 51.20 a | 5.58 a | 5.67 a | 5.58 a |
| | AVG | 14.14 a | 0.520 a | 52.12 a | 5.25 a | 5.50 a | 5.42 a |
| | AVG/GA | 14.05 a | 0.529 a | 52.64 a | 5.17 a | 5.50 a | 5.33 a |
| | AVG/TCG | 14.15 a | 0.526 a | 51.50 a | 5.25 a | 5.50 a | 5.25 a |
| D8 | Control | 16.86 a | 0.409 b | 43.17 b | 6.75 a | 6.50 a | 6.58 a |
| | AVG | 16.03 ab | 0.492 a | 46.18 ab | 5.83 b | 5.58 b | 6.08 ab |
| | AVG/GA | 15.48 b | 0.498 a | 51.15 a | 5.33 b | 5.42 b | 5.42 b |
| | AVG/TCG | 15.71 b | 0.490 a | 49.59 a | 5.75 b | 5.50 b | 5.58 b |
| D12 | Control | 18.83 a | 0.323 b | 36.95 b | 8.17 a | 8.42 a | 8.00 a |
| | AVG | 17.85 ab | 0.396 ab | 40.95 ab | 7.17 ab | 7.33 b | 7.33 ab |
| | AVG/GA | 16.39 c | 0.454 a | 44.87 a | 6.00 b | 6.17 c | 6.25 c |
| | AVG/TCG | 17.10 bc | 0.437 a | 41.49 ab | 6.75 b | 6.60 bc | 6.67 bc |
| D16 | Control | 20.65 a | 0.232 c | 29.58 b | 8.42 a | 8.17 a | 7.83 a |
| | AVG | 19.72 ab | 0.310 b | 34.64 ab | 7.92 a | 7.83 ab | 7.92 a |
| | AVG/GA | 18.61 b | 0.375 a | 39.77 a | 6.67 b | 6.83 c | 7.33 a |
| | AVG/TCG | 18.78 b | 0.355 ab | 36.06 a | 7.58 ab | 7.25 bc | 7.42 a |
| D20 | Control | 21.35 a | 0.180 b | 24.01 c | 6.75 b | 6.83 b | 6.83 b |
| | AVG | 20.53 ab | 0.225 b | 28.88 bc | 7.58 ab | 7.67ab | 7.58 ab |
| | AVG/GA | 19.65 b | 0.298 a | 34.89 a | 8.33 a | 8.33 a | 8.17 a |
| | AVG/TCG | 20.23 ab | 0.243 ab | 31.94 ab | 7.83 a | 8.25 a | 7.83 a |

Means separation within a column for a sampling day by LSD test, where similar letters indicate no significant difference at $p \leq 0.05$; D = sampling day; FW = fresh weight.

TSS is an important factor that determines consumer acceptability for a fresh produce; it generally increases with the storage time due to water loss [17] and the hydrolysis of complex polysaccharides [3]. Reduced TSS accumulation in composite coated persimmon fruits may be due to suppressed metabolic activities and controlled respiration rate, which halted the conversion of starch to soluble sugars [42]. Our findings are confirmed by the previous investigations which reported lower TSS in AVG-based coated fruits such as sapodilla [3], apricot [21] and persimmon [7].

### 3.14.2. Titratable Acidity (TA)

Contrary to TSS, TA decreased in all treatments with the increase in storage time (Table 1). On D20, AVG/GA and AVG/TCG coated persimmon fruits showed higher TA values as compared to AVG coated and non-coated fruits. Organic acids also are a substrate for respiration and thus decrease during fruit ripening [5,25]. The AVG-based composite coatings may modify the internal atmosphere of fruits, limiting physiological processes such as respiration and, thus, retarding the fruit ripening and senescence [21]. Anjum et al. [16] reported higher TA in AVG/GA composite coated guava fruits stored under ambient conditions. Furthermore, higher TA has been reported in AVG-based coated fruits such as papaya [4], sapodilla [3] and persimmon [7].

### 3.14.3. Ascorbic Acid

The reduction in ascorbic acid was non-significant among the treatments during the initial 4 days of storage (Table 1). Overall, AVG/GA maintained a higher amount of ascorbic acid (34.89 mg 100 g$^{-1}$ FW) followed by AVG/TCG (31.94 mg 100 g$^{-1}$ FW) and AVG (28.88 mg 100 g$^{-1}$ FW). Non-coated fruits had the minimum ascorbic acid level (24.01 mg 100 g$^{-1}$ FW) on D20. Ascorbic acid serves as an antioxidant by scavenging free

radicals thereby reducing fruit damage due to oxidation during ripening or in stress conditions, and its amount periodically decreases with time [5,25,41,42]. AVG-based coatings reduced the availability of $O_2$ required for oxidative breakdown, preventing fruit deterioration and senescence. So, AVG-based coating application greatly prevented oxidation of ascorbic acid in harvested fruits [6]. Our results are similar to the findings of other researchers where the application of plant-based edible coatings suppressed the degradation of ascorbic acid in guava [46], apricot [11], persimmon [28] and tomato fruits [18].

### 3.15. Sensory Evaluation

The sensory scores (taste, aroma and overall acceptability) initially increased and then decreased with the passage of time in non-coated persimmon fruits (Table 1). On the other hand, composite coated persimmon fruits exhibited steady development in the sensory scores. On D20, AVG/GA and AVG/TCG coated persimmon fruits showed the highest and statistically similar taste scores (8.33 and 7.83, respectively) followed by AVG coated fruits (7.58). Non-coated persimmon fruits attained maximum aroma and overall acceptability values until D12, and then these reduced with the passage of time. On the contrary, composite coated persimmon fruits had steadily increased aroma scores. On D20, the maximum and statistically similar aroma scores were shown by AVG/GA (8.33) and AVG/TCG (8.25) coated fruits. On D20, overall acceptability scores for different treatments were in decreasing order as follows: AVG/GA > AVG/TCG > AVG > control.

Higher taste scores in non-coated persimmon fruits might be due to higher TSS and lower TA value (Table 1), as sensory attributes are associated with TA, TSS and ripening index [16,34]. High TA and low TSS in coated fruits result in higher sensory attributes for a longer time [13]. In our study, the use of AVG-based composite coatings significantly preserved the biochemical attributes along with delaying the ripening and senescence in persimmon fruits. The early development of colour in persimmon fruits may be due to the carotenoid contents which increased with the passage of time [10]. Reduced accumulation of carotenoids (Figure 2C) in coated persimmon fruits caused the least colour change (Figure 2A,B). So initially, higher overall acceptability in non-coated fruits may be due to early repining in these fruits. Composite coating application suppressed the decay incidence (Figure 1B) in persimmon fruits which is an indicator of customer acceptance. It has been reported that the use of an AVG-based edible coating on sapodilla fruits showed significantly higher sensory scores [3]. Likewise, apricot fruits coated with AVG-based edible coatings conserved high sensory scores during 28 days of storage period [21]. Our results are in accordance with the previous investigations whereby the use of plant-based edible coatings conserved higher sensory scores in coated produces such as apricot [11], guava [16], tomato [18] and mushroom [41].

### 4. Conclusions

The application of AVG-based composite coatings maintained higher fruit quality thereby delaying ripening and senescence. AVG/GA coated persimmon fruits showed reduced physiological weight loss (9.97%), decay incidence (22%), respiration rate (2.89 mmol $CO_2$ kg$^{-1}$ h$^{-1}$) and ethylene production (1.18 µmol kg$^{-1}$ h$^{-1}$). Composite coating application greatly conserved peel colour and total carotenoid contents. AVG/GA coated fruits exhibited lower electrolyte leakage (45.57%), MDA (2.02 µmol kg$^{-1}$), $H_2O_2$ (6.67 µmol kg$^{-1}$ FW) and superoxide anion (2.20 nmol kg$^{-1}$ FW) concentration in comparison with non-coated fruits. Higher activity of enzymatic antioxidants was also recorded in composite coated fruits. AVG/GA coated fruits considerably maintained higher levels of non-enzymatic antioxidants, such as TPC (273.75 mg GAE 100 g$^{-1}$ FW), TFC (158.09 mg QE 100 g$^{-1}$ FW) and soluble tannin (0.08%) than non-coated fruits. AVG/GA coated fruits showed the lower level of water-soluble pectin (7.27 g kg$^{-1}$ FW), and higher level of protopectin (6.06 g kg$^{-1}$ FW), cellulose (79.93 mg kg$^{-1}$ FW) and hemicellulose contents (44.00 mg kg$^{-1}$ FW) than that of non-coated fruits. The use of composite coating

markedly reduced the activity of cell wall degrading enzymes. Reduced TSS and higher TA was also reported in composite coated fruits. Moreover, the application of composite coatings also improved the sensory attributes of coated fruits. Overall, the composite application of AVG and GA could be considered as an appropriate treatment to delay the ripening and maintain the quality of stored persimmon fruits.

**Author Contributions:** Conceptualization, S.E. and M.A.A.; formal analysis, M.S.S., S.U. and H.S.; investigation, M.S.S. and S.U.; methodology, W.F.A.M., M.M.A. and S.A.; supervision, S.E. and M.A.A.; visualization, H.M.A., M.M.A. and A.L.; writing—original draft, M.S.S., S.A., S.E. and H.S.; writing—review and editing, W.F.A.M., H.M.A. and A.L. All authors have read and agreed to the published version of the manuscript.

**Funding:** This publication was funded by the Researchers Supporting Project number (RSP2023R123), King Saud University, Riyadh, Saudi Arabia.

**Data Availability Statement:** Data is contained within the article.

**Acknowledgments:** The authors would like to extend their sincere appreciation to the Researchers Supporting Project number (RSP2023R123), King Saud University, Riyadh, Saudi Arabia. The authors also thank the Department of Horticulture, Bahauddin Zakariya University and Department of Horticulture, MNS-University of Agriculture, Multan, Pakistan for providing laboratory facilities to carry out this research work.

**Conflicts of Interest:** The authors declare no conflict of interest.

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
