# Peer review of "Biocomposite Coatings Delay Senescence in Stored Diospyros kaki Fruits by Regulating Antioxidant Defence Mechanism and Delaying Cell Wall Degradation"

_horticulturae, doi:10.3390/horticulturae9030351_

Round 1

Reviewer 1 Report

Tht authors found that the application of AVG-based edible coatings (AVG/GA and AVG/TCG) maintained quality and delay ripening and senescence in coated persimmon fruits. However, there are still some problems which need to be improved.

(1) Whether the concentrations of the AVG in the three treatments are consistentWhether the Aloe vera gel (AVG) will be diluted in the AVG/GA and AVG/TCGcompare with AVG/

Line 117:AVG/GA and AVG/TCG formulations were prepared by adding AVG solution with either GA or TCG solution in a 1:1 ratio (v/v), respectively.

(2) How about the adhesion on the surface of the fruit? I'm afraid that the gel (adhesive substance) from the AVG will clump together and affect the property of film forming. How did you deal with this problem?

Line 108: The gel matrix was uniformly homogenized in a blender (10,000 × g) to produce a mucilaginous gel and subsequently filtered with a 2-layer muslin cloth.

(3) Line 566: zymes [17]..

(4) Line 623: also inked to ripeness which is influenced by respiration rate

(5) fIgure 7C: the Y-axis.

(6) The results and discussion should be summarized relatively briefly.

(7) Introduction: What is the basis for you selecting GA and TCG, Whether GA or TCG could improve the weaknesses of AVG?

(8) this work? It is not recommended to describe the results of this article here.

Line 75: Moreover, this work also showed that the composite coating mitigated the postharvest oxidative stress by activating the antioxidant defence mechanism and reduced the activity of cell wall degrading enzymes

(9) Please check this method.

The means of treatment were separated by using the least significant difference test (p ≤ 0.05).

Author Response

Dear Editor,

All the queries from the honorable editor and reviewers have been addressed. The changes are highlighted in Green for Reviewer 1 in the revised manuscript.

Reviewer #1:

The authors found that the application of AVG-based edible coatings (AVG/GA and AVG/TCG) maintained quality and delay ripening and senescence in coated persimmon fruits. However, there are still some problems which need to be improved.

(1) Whether the concentrations of the AVG in the three treatments are consistent Whether the Aloe vera gel (AVG) will be diluted in the AVG/GA and AVG/TCG compare with AVG/

Line 117: AVG/GA and AVG/TCG formulations were prepared by adding AVG solution with either GA or TCG solution in a 1:1 ratio (v/v), respectively.

Response: Thanks a lot for your valuable comments. AVG formulation was prepared on the basis of our previously published works as given below.

Saleem et al. (2022) Aloe vera gel coating delays softening and maintains quality of stored persimmon (Diospyros kaki Thunb.) fruits. Journal of Food Science and Technology-Mysore, 59, 3296–3306.

Ali et al. (2020) Effect of pre-storage ascorbic acid and Aloe vera gel coating application on enzymatic browning and quality of lotus root slices. Journal of Food Biochemistry, 44:e13136.

Ali et al.(2019). Aloe vera gel coating delays postharvest browning and maintains quality of harvested litchi fruit. Postharvest Biology and Technology, 157, 110960.

 All finally prepared formulations used for edible coating contained 50% AVG. So, in all coating treatments, AVG concentration was the same (i.e. 50%). This information has been further explained in Section 2.2 and 2.3 for the better understanding of the readers.

(2) How about the adhesion on the surface of the fruit? I'm afraid that the gel (adhesive substance) from the AVG will clump together and affect the property of film forming. How did you deal with this problem?

Line 108: The gel matrix was uniformly homogenized in a blender (10,000 × g) to produce a mucilaginous gel and subsequently filtered with a 2-layer muslin cloth.

Response: We appreciate your expert opinion. The AVG solution was prepared by using the method of (Ebrahimi and Rastegar, 2020). Although pure AVG is too thick to be easily used, diluting it to 50% results in quite useable viscosity, and it forms useable composites with other materials. The blending of the AVG homogenizes all material present in the gel. Moreover, the filtration of the gel removes any remaining clump or other debris. Furthermore, our previously published paper (Saleem et al., 2022) also on AVG-based coating on persimmon fruit.  

Ebrahimi, F.; Rastegar, S. Preservation of Mango Fruit with Guar-Based Edible Coatings Enriched with Spirulina Platensis and Aloe Vera Extract during Storage at Ambient Temperature. Sci. Hortic. (Amsterdam). 2020, 265, doi:10.1016/j.scienta.2020.109258.

Saleem, M.S.; Ejaz, S.; Anjum, M.A.; Ali, S.; Hussain, S.; Nawaz, A.; Naz, S.; Maqbool, M.; Abbas, A.M. Aloe Vera Gel Coating Delays Softening and Maintains Quality of Stored Persimmon (Diospyros Kaki Thunb.) Fruits. J. Food Sci. Technol. 2022, 1–12, doi:10.1007/s13197-022-05412-5.

(3) Line 566: zymes [17]..

Response: Thanks a lot for highlighting the typing error. The sentence has been corrected.

(4) Line 623: also inked to ripeness which is influenced by respiration rate

Response: Thanks a lot for highlighting the typing error. The sentence has been corrected.

(5) Figure 7C: the Y-axis.

Response: Thanks a lot for highlighting the typing error. The error has been corrected in the figure.

(6) The results and discussion should be summarized relatively briefly.

  Response: Thanks a lot for your valuable suggestion. We have modified the results and discussion section in our manuscript as suggested by honorable reviewer.

(7) Introduction: What is the basis for you selecting GA and TCG, Whether GA or TCG could improve the weaknesses of AVG?

Response: Thank you very much for the comments. Although AVG possesses potent antimicrobial and antioxidant properties, but it lacks film-forming properties and, therefore, AVG cannot perform better as compared to other edible coatings with respect to barrier properties (Sarkar et al., 2021). To overcome this issue, film forming compounds such as starch, cellulose, gelatin, gellan gum etc.  are incorporated in the AVG to prepare the composite formulation (Alvarado-González et al., 2012; Ortega-Toro et al., 2017). The structural and chemical interactions in AVG/gellan gum composite film enhanced its functional characteristics, resulting in improved water impenetrability in comparison with AVG alone (Alvarado-González et al., 2012).

Mohebbi et al. (2012) also tested AVG/TCG and found that “…the combination of aloe vera and gum tragacanth was more effective.” than aloe vera or TCG alone.

This has been included in Introduction section for better understanding of the hypothesis L55-77.

Sarker, A.; Grift, T.E. Bioactive Properties and Potential Applications of Aloe Vera Gel Edible Coating on Fresh and Minimally Processed Fruits and Vegetables: A Review. J. Food Meas. Charact. 2021, 15, 2119–2134.

Alvarado-González, J.S.; Chanona-Pérez, J.J.; Welti-Chanes, J.S.; Calderón-Domínguez, G.; Arzate-Vázquez, I.; Pacheco-Alcalá, S.U.; Garibay-Febles, V.; Gutiérrez-López, G.F. Optical, Microstructural, Functional and Nanomechanical Properties of Aloe Vera Gel/Gellan Gum Edible Films. Rev. Mex. Ing. Quim. 2012, 11, 193–202.

Ortega-Toro, R.; Collazo-Bigliardi, S.; Roselló, J.; Santamarina, P.; Chiralt, A. Antifungal Starch-Based Edible Films Containing Aloe Vera. Food Hydrocoll. 2017, 72, 1–10, doi:10.1016/j.foodhyd.2017.05.023.

Mohebbi, M.; Ansarifar, E.; Hasanpour, N.; Amiryousefi, M.R. Suitability of Aloe Vera and Gum Tragacanth as Edible Coatings for Extending the Shelf Life of Button Mushroom. Food Bioprocess Technol. 2012, 5, 3193–3202, doi:10.1007/s11947-011-0709-1.

   (8) This work? It is not recommended to describe the results of this article here.

   Line 75: Moreover, this work also showed that the composite coating mitigated the postharvest oxidative stress by activating the antioxidant defence mechanism and reduced the activity of cell wall degrading enzymes

    Response: Thank you very much for the comments. The sentence has been corrected.

   (9) Please check this method.

   The means of treatment were separated by using the least significant difference test (p ≤ 0.05).

   Response: Thank you very much for the comments. The method has been modified in Section 2.19.

Reviewer 2 Report

For the tittle, the words 'antioxidants' and 'cell wall degradation' are not at the same level. I suggest making the necessary revisions to both words. 

For the results section,

First, in figure 2,the authors measured MDA, leakage, and hydrogen peroxide. I suggest adding the content of superoxide anions. 

Second, figure 3 is about chroma, carotenoids, this part should be next to figure 1 instead of figure 2. To re-organize these figures. 

Third, figure 6 is about activity of cell wall degradation, while figure figure 7 is about the cell wall contents. This logic flue is wrong. I suggest put the cell wall contents first, and then the cell wall degradation enzyme activity, just as the ROS content (figure 2) is before the relevant enzyme activity (figure 4) in the previous figures in this manuscript. 

Fourth, I suggest adding photos of fruits before and after treatment. 

Author Response

Dear Editor,

All the queries from the honorable editor and reviewers have been addressed. The changes are highlighted in Turquoise for Reviewer 2 in the revised manuscript.

Reviewer #2:

(1) For the tittle, the words 'antioxidants' and 'cell wall degradation' are not at the same level. I suggest making the necessary revisions to both words. 

Response: Thanks a lot for highlighting the issue. The title has been modified as suggested by the reviewer.

(2) First, in figure 2the authors measured MDA, leakage, and hydrogen peroxide. I suggest adding the content of superoxide anions.

Response: Thank you for the valuable suggestion. The data regarding superoxide anions content has been added. Further necessary changes regarding superoxide anions have been made in all sections.    

(3) Second, figure 3 is about chroma, carotenoids, this part should be next to figure 1 instead of figure 2. To re-organize these figures. 

Response: Thank a lot, we appreciate your comments. We have rearranged the figures according to kind suggestions of the reviewer.

(4) Third, figure 6 is about activity of cell wall degradation, while figure figure 7 is about the cell wall contents. This logic flue is wrong. I suggest put the cell wall contents first, and then the cell wall degradation enzyme activity, just as the ROS content (figure 2) is before the relevant enzyme activity (figure 4) in the previous figures in this manuscript. 

Response: Thank a lot, we appreciate your comments. We have rearranged the figures according to kind suggestions of the reviewer.

(5) Fourth, I suggest adding photos of fruits before and after treatment. 

Response: Thank you very much for the comments. The photos of the fruits— before storage and after storage time completed— haves been added  in the manuscript (Figure 2D).

Reviewer 3 Report

For some devices, the type, manufacturer and where it was manufactured are not listed, so this needs to be corrected.

The whole experiment is well planned and also well described in the methods.

The introduction is too general, so it is necessary to write only what concerns the topic of scientific research.

The conclusion should be corrected and supported by significant results (numerically).

Author Response

Dear Editor,

All the queries from the honorable editor and reviewers have been addressed. The changes are highlighted in Pink for Reviewer 3 in the revised manuscript.

Reviewer #3:

(1) For some devices, the type, manufacturer and where it was manufactured are not listed, so this needs to be corrected.

Response: Thanks a lot for the clarification. The name, device type and necessary information have been added in Section 2.

(2) The whole experiment is well planned and also well described in the methods.

Response: Thank you very much for your positive comments.

(3) The introduction is too general, so it is necessary to write only what concerns the topic of scientific research.

Response: Thank you for the suggestion. We have tried our best modify the introduction according to your kind suggestions.

(4) The conclusion should be corrected and supported by significant results (numerically).

Response: Thanks a lot for your constructive comments. We have modified the Conclusion section according to the suggestion of honorable Editor and Reviewer.

Round 2

Reviewer 2 Report

I feel most of my concerns have been addressed by the authors, I have no further comments.